# The Neuroinflammatory Role of Pericytes in Epilepsy

**DOI:** 10.3390/biomedicines9070759

**Published:** 2021-06-30

**Authors:** Gaku Yamanaka, Fuyuko Takata, Yasufumi Kataoka, Kanako Kanou, Shinichiro Morichi, Shinya Dohgu, Hisashi Kawashima

**Affiliations:** 1Department of Pediatrics and Adolescent Medicine, Tokyo Medical University, Tokyo 160-8402, Japan; kanako.hayashi.0110@gmail.com (K.K.); s.morichi@gmail.com (S.M.); hisashi@tokyo-med.ac.jp (H.K.); 2Department of Pharmaceutical Care and Health Sciences, Faculty of Pharmaceutical Sciences, Fukuoka University, Fukuoka 814-0180, Japan; ftakata@fukuoka-u.ac.jp (F.T.); ykataoka@fukuoka-u.ac.jp (Y.K.); dohgu@fukuoka-u.ac.jp (S.D.)

**Keywords:** pericytes, mural cells, cytokine, blood-brain barrier, neuroinflammation

## Abstract

Pericytes are a component of the blood–brain barrier (BBB) neurovascular unit, in which they play a crucial role in BBB integrity and are also implicated in neuroinflammation. The association between pericytes, BBB dysfunction, and the pathophysiology of epilepsy has been investigated, and links between epilepsy and pericytes have been identified. Here, we review current knowledge about the role of pericytes in epilepsy. Clinical evidence has shown an accumulation of pericytes with altered morphology in the cerebral vascular territories of patients with intractable epilepsy. In vitro, proinflammatory cytokines, including IL-1β, TNFα, and IL-6, cause morphological changes in human-derived pericytes, where IL-6 leads to cell damage. Experimental studies using epileptic animal models have shown that cerebrovascular pericytes undergo redistribution and remodeling, potentially contributing to BBB permeability. These series of pericyte-related modifications are promoted by proinflammatory cytokines, of which the most pronounced alterations are caused by IL-1β, a cytokine involved in the pathogenesis of epilepsy. Furthermore, the pericyte-glial scarring process in leaky capillaries was detected in the hippocampus during seizure progression. In addition, pericytes respond more sensitively to proinflammatory cytokines than microglia and can also activate microglia. Thus, pericytes may function as sensors of the inflammatory response. Finally, both in vitro and in vivo studies have highlighted the potential of pericytes as a therapeutic target for seizure disorders.

## 1. Introduction

Accumulating evidence has demonstrated that the pathogenesis of epilepsy is linked to neuroinflammation and cerebrovascular dysfunction [1,2,3,4,5,6]. Traditionally, microglia had been considered to be responsible for the cytokine-centered immune response in the central nervous system (CNS); however, brain pericytes can respond to inflammatory signals, such as circulating cytokines, and convey this information to surrounding cells through chemokine and cytokine secretions [7,8,9,10]. Recent studies have demonstrated that pericytes may act as sensors for the inflammatory response in the CNS, as pericytes react intensely to proinflammatory cytokines when compared to other cell types (e.g., microglia) that constitute the CNS and factor-induced reactive pericytes can also activate microglia in vitro [9,11,12,13].

Pericytes provide physical support to the blood–brain barrier (BBB) and play an integral role in CNS homeostasis and BBB function [14]. Pericyte degeneration and/or dysfunction contribute to the loss of BBB integrity, which is an early hallmark of several neurodegenerative and inflammatory conditions [8,15,16]. Another notable feature of pericytes is their ability to regulate the migration of leukocytes across the brain microvascular endothelial cell (BMVEC) barrier, which secretes key molecules that support the BBB barrier [17,18]. Recent research on the pathogenesis of epilepsy has begun to elucidate the mechanisms mediating peripheral-to-CNS cell infiltration in human and mouse models [19,20]. Pericytes may contribute to the mechanisms, while emerging research is investigating the extent of peripheral immune cell involvement in the inflammatory pathology of epilepsy.

The various functions of pericytes and their involvement in CNS diseases, including ischemic stroke [21], spinal cord injury [22], brain injury [23], and multiple sclerosis [24], has been reported.

The association between pericytes and epilepsy has attracted attention, while several recent studies have illustrated the contributions of pericytes to the pathogenesis of epilepsy [2,25,26,27,28,29,30,31,32]. These studies suggested that pericytes might participate in the pathogenesis of epilepsy, consisting of neuroinflammation and BBB damage and the interaction between peripheral and central immunity. Thus, evidence on the relationship between pericytes and the pathogenesis of epilepsy is gradually accumulating. Therefore, this study aimed to investigate the pathogenesis of epilepsy and pericytes because none of the review articles focused on this, even though therapeutic targets for pericytes in neurological disorders were investigated [17,33,34].

This review (1) explores the current literature regarding the role of pericytes in the pathogenesis of epilepsy and (2) highlights novel directions for research on therapeutic interventions for epilepsy that target pericytes. Given the paucity of knowledge on pericyte function in seizures and epilepsy-related pathologies, further studies are warranted to investigate pericytes as a potential therapeutic target for epilepsy treatment.

## 2. What Are Pericytes?

Pericytes were first described by the French scientist Charles-Marie Benjamin Rouget and were originally called Rouget cells in 1873 [35]. Later, this population was rediscovered by Zimmermann as a cell that shows a specific morphology around microvessels, and became widely known as a “pericyte” [36]. Pericytes are mural cells that are implanted in the basal membrane surrounding endothelial cells in capillaries and small vessels, including precapillary arterioles and postcapillary venules. Although the origin of all pericytes has not been clarified, blood vessels in the CNS are predominantly covered by neural crest cell-derived pericytes, while mesoderm-derived pericytes mainly contribute to blood vessel coverage in the trunk [37]. In the brain, pericytes constitute a vital component of the BBB/neurovascular unit (NVU) and cover the BMVECs lining the capillaries on the parenchymal side, where there are astrocytic end feet that enclose cerebral vessels, perivascular microglia/macrophages, and neurons [17,38,39]. Pericytes form a crucial component of the brain microvasculature and play an integral role in CNS homeostasis and BBB function [14] in normal physiological (Figure 1) and pathological conditions (Figure 2). A potential mechanism of pericyte action is the regulation of signaling through platelet-derived growth factor receptor beta (PDGFRβ), which is commonly used as a marker of pericytes and regulates pericyte survival, proliferation, and migration signals [40]. In the CNS, platelet-derived growth factor-beta subunit (PDGF-BB) is released by endothelial cells and binds to PDGFRβ at the cell surface of pericytes to promote pericyte vascularization within the BBB [41]. The PDGFRβ signaling pathway is involved in pericyte survival and subsequent development as well as the function of the BBB during adulthood and senescence, as demonstrated by experiments in pericyte-deficient mice [17,38]. In addition to its role as a marker of CNS pericytes, PDGFRβ is expressed in oligodendrocyte precursor cell (OPC)/neuron-glial antigen 2 (NG2) parenchymal glial cells [2,25,42]. Other markers for pericytes exist (Table 1), but these remain inconclusive. Anatomical studies are required to investigate the characteristics of pericytes that possess longitudinal processes along vessels and contribute to BBB maintenance [15]. Pericytes in the brain are highly heterogeneous and have different morphologies as well as functions depending on their location in the vasculature [10]. Further, transgenic mice generated to study pericyte function may yield information on other cell types [43]. Therefore, “peripheral blood-specific” markers must be used with caution [44]. Although there is no scientific consensus on what constitutes true pericytes [45], the current review focuses on studies using definitive pericyte-related markers and anatomy.

## 3. Pericytes and Neuroinflammation

Evidence accumulated from experimental models and human samples implicates immunological processes in the pathogenesis of epilepsy [1,4]. The involvement of pericytes in the CNS immune responses has attracted significant attention. Pericytes present heterogeneous signals to the surrounding cells and actively modulate inflammatory responses in a tissue- and context-dependent manner. The expression of various pattern-recognition receptors (PRRs), including toll-like receptors (TLRs) and nucleotide-binding and oligomerization domain (NOD)-like receptor families, has been detected in brain pericytes [52]. Given the abundance of surface receptors, pericytes can respond to inflammatory mediators, such as monocyte chemoattractant protein-1 (MCP-1/CCL2) and tumor necrosis factor (TNF)-α, which in turn induce the secretion of CCL2, nitric oxide (NO), and several cytokines [7,8,9,53]. Pericytes act as promoters of both the innate and adaptive immune system [43]. In the CNS, microglia are a hallmark of the immune response, which produce cytokines such as interleukin (IL)-1β, TNF-α, IL-6, and various other chemokines [54], and related effector pathways, including cyclooxygenase-2 (COX-2)/prostaglandin (PGE2) and complement factors [55]. The rapid activation of microglia impairs neuronal function by inducing inflammatory mediators, such as NO, reactive oxygen species (ROS), and proinflammatory cytokines [56,57].

Pericytes have been shown to be more sensitive to proinflammatory cytokines compared to other cells in the NVU [9,11,12,13]. Specifically, cytokine and chemokine release profiles from brain pericytes in response to TNF-α are distinct to those of other cell types comprising the NVU, and TNF-α-stimulated pericytes release macrophage inflammatory protein (MIP)-1α and IL-6. Among BBB cells, pericytes stimulated with TNF-α induced the highest levels of *iNOS* and IL-1β mRNA expression, which indicates the activation of BV-2 microglia [9]. The mechanism underlying TNF-α-induced IL-6 release involves the inhibitor kappa B (IκB)-nuclear factor kappa-light-chain-enhancer of activated B cells (NFκB) and the Janus family of tyrosine kinase (JAK)-signal transducer and activator of transcription (STAT) 3 pathways [13]. NFκB plays a key role in inflammation, immune, and stress-related responses, as well as in the regulation of cell survival and in the growth of neural processes in developing peripheral and central neurons [58]. These findings indicate that the activated brain pericytes trigger the development of uncoordinated NVU function, including glial activation, and may act as sensors at the BBB in TNF-α-mediated brain inflammation.

Pericytes also release anti-inflammatory factors, highlighting their involvement in regeneration and protection [7,59,60]. Pericytes respond to lipopolysaccharide (LPS), secrete anti-inflammatory cytokines such as IL-10 and IL-13 [61], and produce neurotrophins such as nerve growth factor (NGF) and brain-derived neurotrophic factor (BDNF), which regulate neuronal development [42,62]. Pericytes upregulate neurotrophin-3 production in response to hypoxia, resulting in increased NGF production in astrocytes, thereby protecting neurons from hypoxia-induced apoptosis [62]. These actions highlight the neuroprotective functions of pericytes under pathological conditions.

## 4. Pericytes and Epilepsy

Table 2 summarizes the research on pericytes and epilepsy.

## 5. Blood-Brain Barrier Disruption in the Pathogenesis of Epilepsy

Experimental evidence of BBB impairment in the pathogenesis of epilepsy has been demonstrated in patients and animal models [64,65,66,67], which is a hallmark of epilepsy. BBB disruption can also directly induce seizure activity and exacerbate epileptogenesis; the relationship between epilepsy and BBB breakdown is bidirectional [64,65].

BBB dysfunction and subsequent infiltration of serum albumin into the brain leads to changes in epileptogenesis, including astrocyte changes, neuroinflammation, excitatory synapse formation, and pathological plasticity [68,69]. These BBB alterations are not only due to leakage, as demonstrated by Evans Blue staining [65]. There is involvement of various inflammatory mediators as nondisruptive changes at the molecular level of pericytes are also involved in the changes of the BBB; specifically, they secrete various mediators as follows: IL-1β, TNF-α, IFN-γ, matrix metalloproteinases (MMPs), ROS/reactive nitrogen species (RNS), (NO), and prostaglandin E2 (PGE2). Pericyte-derived MMP-9 upregulation in the cerebral microvasculature can cause endothelial dysfunction through degradation of tight junctions and extracellular matrices, resulting in subsequent pericyte loss from the microvasculature and BBB disruption [11,43]. Moreover, the secretion of ROS/RNS, NO, and PGE2 lead to vasodilation and breaching of the BBB [9]. Epileptic seizures can cause pericytes surrounding the blood vessels to rearrange [2] and morphologically alter, which is facilitated by the inflammatory mediators [29,30]. These series of alterations are thought to be linked to the pathogenesis of epilepsy, although further details are warranted.

## 6. Leukocyte Recruitment and Peripheral-to-Central Infiltration

Pericytes regulate the migration of leukocytes across the BMVEC barrier and secrete key molecules that support the BBB [17,18]. Chemokines (CCL2, CXCL1, CXCL8, and CXCL10) secreted by pericytes in both basal and inflammatory states recruit peripheral immune cells, including monocytes, B and T cells, and neutrophils, to the CNS parenchyma via upregulation of intercellular adhesion molecule-1 (ICAM-1) and vascular cell adhesion molecule-1 (VCAM-1) on the endothelium [7,8,9,70]. Although the human brain is considered an immune-privileged area [68,71], this is not preserved during inflammatory conditions. Analysis of brain parenchyma in patients with epilepsy showed that there have been both positive [72,73] and negative [74] reports on the occurrence of infiltration of peripheral leukocytes into the brain tissue. Recent experimental research demonstrated that peripheral-to-CNS cell infiltration, particularly monocytes, occurs in the status epilepticus (SE) model, without evidence of infections or immune disorders [20,75,76]. The possibility of classifying peripheral monocytes and indigenous microglia, which have been considered difficult to differentiate, has been increased using genetic engineering [75,77,78].

In chemokine receptor 2 (CCR2)-knockout mice, the CCL2 receptor, which blocks peripheral monocyte invasion into the brain tissue, attenuated neuronal damage in SE models [75]. Analysis of the brain tissue from pediatric patients with drug-resistant epilepsy (DRE) revealed that seizure frequency was correlated with the number of infiltrating peripherally activated CD3+ T cells and monocytes, but not microglia [19]. Current analysis of pediatric patients with DRE also demonstrated a correlation between the number of seizures and intracellular IL-1β levels in monocytes [79], while experimental data and human research attributed seizure-induced neuronal death to the activation of resident microglia [78,80]. Whether the peripheral monocytes or the resident microglia are the primary triggers of epilepsy, as well as the extent to which the infiltrated cells are significant, remains to be determined; nevertheless, the combination of the roles of the pericytes in maintaining the BBB integrity, producing inflammatory mediators, and recruiting leukocytes indicate that the pericytes could be intimately involved in the pathogenesis of epilepsy.

## 7. Clinical Evidence Links Pericytes to Epilepsy

The disarray of the pericyte-basal lamina interface in patients with epilepsy was first described in 1990 [63]. Evidence of pericyte degeneration with basement membrane unit thickness and cytoplasmic density has also been reported in most of the spiking area microvessels in human brain tissues of intractable complex partial seizures using an electron microscope [63].

With the advent of PDGFRβ, though a nonspecific CNS pericyte marker, the immunostaining reports of the presence of PDGFRβ+ cells have emerged in the brain specimens of patients with intractable epilepsy in focal cortical dysplasia (FCD) and temporal lobe seizures (TLE) [2,25,29]. In tissues from patients with refractory TLE and hippocampal sclerosis (HS), the presence of PDGFRβ+ cells associated with blood vessels and parenchyma was observed, although findings were heterogenous [2]. Indeed, the highest perivascular PDGFRβ immunoreactivity was detected in patients with TLE-HS, specifically in the microvasculature [2]. Tissue from patients with cryptogenic epilepsy has exhibited a similar immune response pattern, although to a lesser extent than that of FCD. Increased perivascular PDGFRβ immunoreactivity was associated with increased hippocampal vascularization in the cells of patients with TLE-HS [25].

Another study of TLE and FCD specimens revealed robust PDGFRβ-positive cell pericyte immunoreactivity surrounding the blood vessels, particularly in TLE with HS specimens, with aggregation of IBA1/HLA microglial cells and pericyte-microglia outlining the capillary wall [29]. The morphological changes in pericytes were induced by proinflammatory cytokines, including IL-1β, TNFα, and IL-6; in particular, IL-6 exposure was drastically associated with apoptosis, suggesting pericyte damage [29].

Collectively, the accumulation of pericytes (PDGFRβ-positive cells) in the cerebral vascular regions was consistently observed in patients with refractory epilepsy [2,25,29]. The degree of accumulation correlates to some extent with the clinical picture [25,29], and morphological changes of the pericytes might be due to proinflammatory cytokines [29]. In addition, the amount of angiogenesis, which is associated with epileptogenesis, was related to the number of PDGFRβ-positive cells [25], suggesting a relationship between PDGFRβ-positive cells and the pathogenesis of epilepsy.

## 8. Experimental Evidence Links Pericytes to Epilepsy

An in vivo study of NG2DsRed mice, which enabled the visualization of cerebrovascular pericytes, revealed heterogeneous perivascular prominence of NG2DsRed cells with PDGFRβ expression in an SE model induced by intraperitoneal kainic acid (KA) [2]. These heterogeneous perivascular patterns of PDGFRβ+ cells are inconsistent with the aforementioned human tissue findings [2,25,29], which have also been observed in a rat model of neurovascular dysplasia SE, particularly in the hippocampus with a neurovascular dysplasia SE rat model [25].

An in vitro and in vivo study by Milesi et al. demonstrated that the parenchymal and vascular PDGFRβ+ cells were redistributed, alongside partial colocalization of vascular and parenchymal PDGFRβ+ cells with NG2DsRed and NG2, but not with IBA-1 [2]. These findings, suggesting that the accumulation of pericytes and microglia is associated with epileptic seizure events, have been documented in recent studies [29,30].

Klement et al. employed a model of TLE (associated with HS) in NG2DsRed mice to assess the impact of seizure progression on capillary pericytes and surrounding glial cells [29]. In vivo, SE mice presenting with spontaneous recurrent seizures (SRS) exhibited disorganized NG2DsRed-positive pericyte somata in the hippocampus at 72 h and 1 week after SE (epileptogenesis) in the hippocampus. Pericyte modifications clustered with IBA1-positive microglia, surrounding capillaries, and overlapped topographically with pericytes lodged within microglial cells [29]. Residual microglial clustering was also observed surrounding NG2DsRed pericytes in SRS, proinflammatory mediators, such as IL-1β, IL-6, TNF-α, and particularly IL-1β; however, the in vitro study in humans revealed that IL-6 induced these morphological changes of pericyte-microglia clustering in NG2DsRed hippocampal slices [29]. In addition, Klement et al. also reported a pericyte-glia perivascular scar with capillary leaks in the hippocampus during seizure activity. These scars in the cornu ammonis region developed an abnormal distribution or accumulation of extracellular matrix collagen III/IV as the seizure progressed [30]. In vitro experiments induced by 4-aminopyridine and low-Mg^2+^ conditions repeated seizures that cause vasoconstriction associated with the depolarization of mitochondria in pericytes and gradual neurovascular disconnection, suggesting that the pericyte damage causes vascular dysfunction in epilepsy [31]. The gradual progression of neurovascular decoupling during recurrent seizures suggests that pericyte damage induces vascular dysfunction in epilepsy (Figure 3) [31].

## 9. Prospects for Pericyte-Mediated Epilepsy Therapy

PDGFRβ can regulate pericyte survival, proliferation, and migration signals and is commonly used as a marker for pericytes [40]; PDGFRβ suppression has been proposed as a possible treatment for epilepsy [28,30,32].

As described above, a pericyte-glia perivascular scar with capillary leaks induced by seizures and a high expression of PDGFRβ transcript and protein levels were detected [30]. In the organotypic hippocampal cultures, PDGFRβ reactivity surrounding capillaries is also enhanced by electrographic activity and was reduced by PDGF-BB (a PDGFRβ agonist) and PDGFβ inhibitor imatinib [30]. Furthermore, PDGF-BB can reduce mural cell loss, vascular pathology, and epileptiform electroencephalography activity in a KA-induced SE model [28]. Recently, traumatic brain injury (TBI) has been highlighted as a major factor in epilepsy owing to certain intractable cases. The evaluation of the involvement of pericytes in the pathogenesis of epilepsy was performed using a controlled cortical impact (CCI) device. PDGFRβ levels were significantly increased following CCI in the injured ipsilateral hippocampus; pilocarpine-induced seizures can be regulated by imatinib treatment in this CCI model [32]. The efficacy of imatinib was also observed in vitro.

The findings from both in vitro and in vivo studies highlight the potential of pericytes as a therapeutic target for seizure disorders, as indicated by the efficacy of PDGF-BB and imatinib in blocking PDGFRβ. However, both PDGFRβ and PDGF-BB are required for the pericyte coating of the BBB in the developing CNS [38,41]. Under pathological conditions, mural cells in the immediate postacute phase (SE, ischemic stroke, and head trauma) require support from the PDGFRβ activation [28]; hence, the inflammatory involvement of PDGFRβ may be relevant in long-term progression as well as in chronic stages.

When considering the pharmacological modulation of pericyte signaling pathways as a means of attenuating disease progression and capillary pathology, the impact of pericyte modulation in the epileptic brain must consider the activation state of the glial cells and the disease stage (e.g., acute vs. chronic) [29]. Further, considering the distinct functions of PDGFRβ at different developmental stages, the timing of PDGFRβ inhibition needs to be carefully studied; moreover, avoiding imatinib in the acute phase of the disease may be considered. It remains debatable whether the changes in pericytes and accumulation of microglia associated with PDGFR expression in this series of studies should be suppressed.

Transforming growth factor-beta 1 (TGFβ1) is a multifaceted cytokine in the brain that plays a role in regulating cell proliferation, differentiation, survival, and scar formation [82,83]. Since 1989, the possibility of PDGF-induced TGF-β signaling has been suggested [84]; PDGFR-β and TGF-β with PDGFR-β might mediate the endothelial cell/pericyte interaction to protect the BBB integrity [33]. The potential involvement of TGF-β in epileptogenesis has been recognized from an experimental model showing TGF-β upregulation as part of the inflammatory response [85]. Microarray analysis of TGFβ1-stimulated human brain pericytes isolated from intractable TLE demonstrated inhibition of pericyte proliferation and phagocytosis by TGFβ1 [27]. However, TGFβ1 also enhanced the expression of IL-6, MMP-2, and NOX4, which can disrupt BBB functioning; thus, these reactions caused by TGFβ1 might not lead to the treatment of the neurovascular system [27].

Although the brain pericyte-derived TGF-β contributes to the upregulation of BBB functions [86], suppression of TGFβ1 indicates improvement in epilepsy [87]. Losartan, an angiotensin-type 1 receptor (AT1) antagonist, prevents phosphorylation of Smad proteins of TGF-β signaling [88,89], which has demonstrated both neuroprotective and antineuroinflammatory effects [90,91,92].

These in vitro studies also suggest that human-derived pericytes are morphologically altered by proinflammatory cytokines that induce apoptosis [29], indicating the potential of targeting IFN-γ for pericyte-mediated epilepsy treatment [26]. IFN-γ is a central component of the CNS inflammatory response and is secreted by microglia, astrocytes, endothelial cells, and circulating immune cells [93,94,95]. This classical inflammatory mediator has been implicated in CNS diseases, including epilepsy [96,97]. Altering the proportion of microglial phenotypes via IFN-γ treatment improved the prognosis in a mouse model of epilepsy [98].

Notably, in epileptiform conditions, IL-1β, a neurotoxic cytokine and one of the cytokines chiefly involved in the pathogenesis of epilepsy, prominently contributes to the morphological changes in the pericytes [29]. There is evidence that the IL-1/IL-1R1 axis plays an important role in the inflammatory response in epilepsy, as presented by Vezzani et al. in an excellent review [4,99]. IL-1β agonist, the IL-1 receptor antagonist (IL-1RA), has already been tested for clinical application for epileptic syndromes using anakinra, and has shown favorable clinical outcomes [100,101,102,103]. The use of anakinra on pericytes in status epilepticus has not yet been investigated. To ensure the involvement of pericytes in epilepsy, it is worthwhile to confirm that anakinra suppresses the morphological changes in pericytes and reduces seizures.

Previous reports have demonstrated that inhibition of pericytes could have positive effects of neuroprotection [26,28,30,32]; however, there is also a concern that the suppression of pericytes by TGFβ1 may not necessarily have a positive effect on the CNS [27]. Since TGFβ1 suppresses pericyte phagocytosis and reduces the expression of central leukocyte trafficking chemokines and adhesion molecules while increasing the expression of proinflammatory cytokines and enzymes that promote BBB disruption, a paradoxical reaction has been reported [27]. The TGFβ1 response of pericytes may differ from the anti-inflammatory response of microglia [104,105,106,107]; therefore, further studies are required to obtain any effect on this nonuniform response.

In the pathogenesis of epilepsy, pericytes adopt a phenotype that is neither solely pro- nor anti-inflammatory [27]. Merely suppressing pericytes may not be sufficient to improve the treatment of epilepsy, and it may be necessary to seek a treatment tailored to the affected child in combination with various therapies that have been introduced in recent reviews [108].

## 10. Conclusions

In this review, we present evidence for the substantive role of pericytes in the pathogenesis of epilepsy. The roles of pericytes in maintaining BBB integrity, producing inflammatory secretions, and recruiting leukocytes highlights the potential role of pericytes in the pathogenesis of epilepsy. Pericytes may also act as sensors of inflammatory processes in the CNS and regulating them may lead to the development of novel therapies for epilepsy. However, as there remains a lack of absolute molecular markers for pericytes, and since pericytes originate from multiple cellular sources and vary in morphology, localization as well as function in different tissues leaves several issues to be addressed. In addition, we are unable to determine whether brain inflammation is an initiator or a consequence of a systemic inflammatory process.

Several reports have suggested entry points that may also act as a basis for various neurovascular therapies, including anakinra [100,101] and losartan [87], though the level of evidence for both drugs is limited for the establishment of treatment for epilepsy. These drugs provide an avenue for novel therapeutic, anti-inflammatory, or cerebrovascular repair to mitigate epileptic pathophysiology. Unfortunately, definitive treatments for epilepsy are currently lacking. BBB integrity and systemic peripheral inflammation may contribute to epilepsy and hold potential for molecular biomarkers and targets in the treatment of epilepsy. Moreover, human pluripotent stem cell-derived brain pericyte-like cells induced BBB properties in BMECs, resulting in strengthening of the barrier and a reduction in transcytosis [109]. These stem cell techniques could be applied to examine the possibility of new strategies to selectively target pericytes and the role of pericytes in epilepsy more specifically. Novel tools to control pericytes should be developed to target inflammatory vascular-related processes during seizure progression or activity.

## Figures and Tables

**Figure 1 biomedicines-09-00759-f001:**
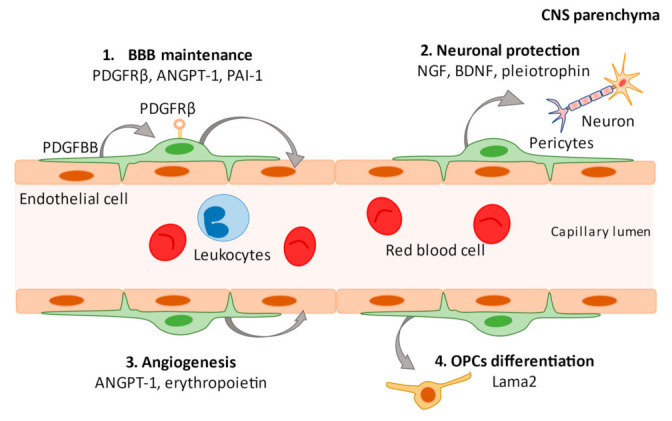
Regulatory functions of pericytes. In the central nervous system (CNS), platelet-derived growth factor-beta subunit (PDGF-BB) is released by endothelial cells and binds to PDGFRβ at the cell surface of pericytes to promote pericyte vascularization within the blood–brain barrier (BBB). Secretion of angiopoietin-1 (ANGPT-1) and plasminogen activator inhibitor type 1 (PAI-1) from pericytes promotes the development of vascular endothelial cells and contributes to the maintenance of the BBB (1). Pericytes maintain neuronal health by secreting factors such as nerve growth factor (NGF), brain-derived nerve growth factor (BDNF), and pleiotrophin (2). Pericytes are involved in angiogenesis by secreting ANGPT-1 and erythropoietin (3) and produce a factor (Lama2) that facilitates the differentiation of oligodendrocyte progenitor cells (OPCs) into mature oligodendrocytes (4).

**Figure 2 biomedicines-09-00759-f002:**
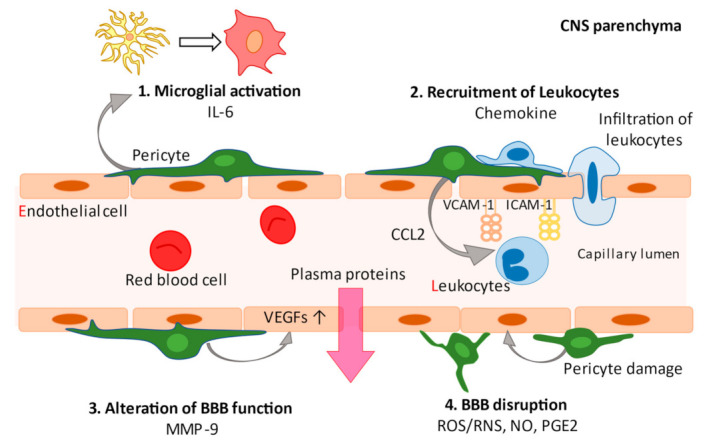
In pathological conditions, pericytes generate various inflammatory factors. Pericytes secrete IL-6 that can polarize parenchymal microglia to a proinflammatory phenotype to activate microglia (1). The secretion of chemokines (CCL2, CXCL1, CXCL8, and CXCL10) by pericytes recruits leukocytes to the CNS parenchyma via the upregulation of ICAM-1 and VCAM-1 adhesion molecules on the endothelium (2). MMP-9 secretion stimulates the production and secretion of vascular endothelial growth factor (VEGF), resulting in endothelial dysfunction (3). Secretion of reactive oxygen species/reactive nitrogen species (ROS/RNS), nitric oxide (NO), and prostaglandins (PGE2) by pericytes lead to vasodilation and breaching of the blood–brain barrier. Pericytes themselves are morphologically altered by inflammatory mediators (4).

**Figure 3 biomedicines-09-00759-f003:**
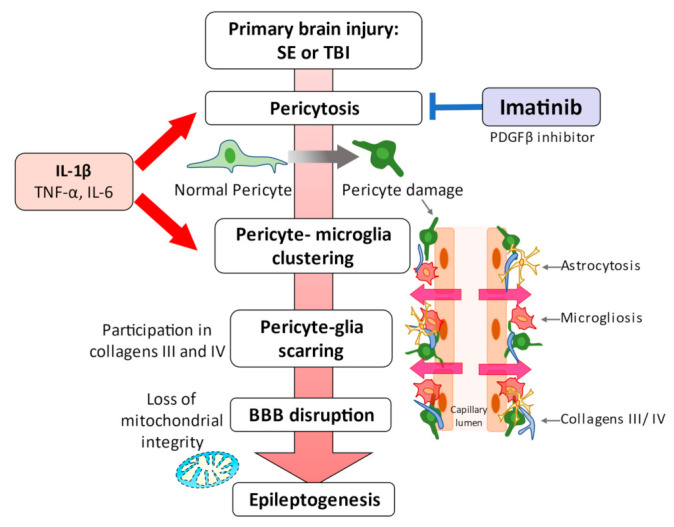
Schematic representation of the events linking pericytes to epilepsy. Status epilepticus leads to redistribution and remodeling of cerebrovascular pericytes, potentially contributing to blood–brain barrier permeability [2,28,29]. A significant clustering of microglia/macrophages around pericytes occurs one week after the attack, although pericyte proliferation is significantly increased as early as 72 h [29]. These series of pericyte-related modifications are promoted by proinflammatory cytokines, including IL-1β, TNFα, and IL-6. Alterations caused by IL-1β, which is one of the cytokines most deeply involved in the pathogenesis of epilepsy, were most pronounced. These pericyte-associated modifications and pericyte-microglia clustering may be facilitated by IL-1β [29], and pericyte-glial scarring with collagens III and IV process leaky capillaries during seizure progression [30]. Recurrent seizures can lead to pericytic injury with neurovascular decoupling and BBB dysfunction at the arterial and capillary levels. Moreover, capillary vasoconstriction is accompanied by a loss of mitochondrial integrity in pericytes [81]. In vitro and in vivo studies have highlighted the potential of pericytes as a therapeutic target for seizure disorders [28,30,32].

**Table 1 biomedicines-09-00759-t001:** Common markers used to identify pericytes in the central nervous system of mice that also label other cell types.

Marker	Cells Labeled	Main Function	Reference(s)
PDGFRβ (platelet-derived growth factor receptor beta)	Fibroblasts, SMCs, pericytes	Tyrosine kinase receptor	[14,41]
NG2 (CSPG4; chondroitin sulfate proteoglycan 4)	OPCs, NSCs, SMCs, pericytes	Cell-membrane proteoglycan	[46]
CD13 (aminopeptidase N)	Fibroblasts, SMCs, pericytes	Cell-membrane aminopeptidase	[14]
αSMA (actin, aortic smooth muscle)	SMCs, myofibroblasts, pericytes	Cytoskeletal protein	[14]
Desmin	SMCs, pericytes	Intermediate filament	[14]
Rgs5 (regulator of G protein signaling 5)	SMCs, pericytes	Regulator of G protein	[47]
CD146 (cell surface glycoprotein MUC18)	SMCs, pericytes	Membrane proteins	[48]
SUR2 (sulfonylurea receptor 2)	SMCs, pericytes	Potassium-channel	[47,49]
Kir6.1 (K+ channel pore-forming subunit)	SMCs, fibroblasts, pericytes	Potassium-channel	[47,49]
NeuroTrace 500/525 (fluorescent Nissl dye/FluoroNissl Green)	Pericytes	-	[50]
Vitronectin	SMCs, Pericytes	Complement-binding protein	[49,51]

Note: NSCs, neural stem cells; OPCs, oligodendrocyte progenitor cells; SMCs, smooth muscle cells.

**Table 2 biomedicines-09-00759-t002:** Research and key findings on pericytes and epilepsy.

No.	Patients/Model	Species	Key Findings	Reference
1	Intractable complex partial seizures	Humans	Degeneration of pericytes (aggregates of cellular debris within the basement membrane) with the morphological changes in pericyte-basement membrane unit thickness and pericyte cytoplasmic density were observed in the spiking area of microvessels in an electron microscopy study of brain tissue	[63]
2	TLE with HS	Humans	PDGFRβ+ cells are distributed around the cerebrovasculature and are present in the brain parenchyma of human TLE specimens	[2]
	NG2DsRed or C57BL/6J mice (intraperitoneal KA injections)	Mice	Constitutive cerebrovascular NG2DsRed pericyte coverage is impaired in response to SE in vivo or seizure-like events in vitro Redistribution of parenchymal and vascular PDGFRβ+ cells occurs in vitro and in vivo Vascular and parenchymal PDGFRβ+ cells partially co-localize with NG2DsRed and NG2, but not with IBA-1 (indicators of microglia)	
3	FCD, TLE without HS, cryptogenic epilepsy	Humans	FCD and TLE-HS display the highest PDGFRβ immunoreactivity at the microvasculature identifying pericytes Cryptogenic epilepsy patients also showed a similar immune response pattern, although to a lesser extent than that in FCD The amount of perivascular PDGFRβ immunoreactivity was found to be associated with increased hippocampal angiogenesis in tissues from patients with TLE-HS	[25]
	Neurovascular dysplasia rat model (Sprague-Dawley rats with pre-natal exposure to methyl-axozy methanoic acid), pilocarpine	Mice	Pericyte-vascular dysplasia was detected in hippocampi corresponding to neuronal heterotopias Severe SE was associated with a region-specific increase in PDGFRβ immunoreactivity	
4	TLE	Humans	Chronic IFN-γ treatment blocks signaling through PDGFRβ by enhancing agonist PDGF-BB	[26]
5	Drug-resistant TLE (microarray analysis)	human	TGFβ1 decreased pericyte proliferation and decreased phagocytosis TGFβ1 also upregulates the expression of IL-6, MMP-2, and NOX4, which disrupt the function of the BBB, and these responses to TGFβ1 may not be therapeutic for the neurovascular system	[27]
6	Dynamics of NG2 mural cells under SE with systemic KA injection in mice	Mice	NG2 mural cells are added and removed from veins, arterioles, and capillaries after status epilepticus Loss of NG2 mural cells is proportional to seizure severity and vascular pathology (e.g., rigidity, perfusion, and permeability) Treatment with PDGF-BB reduced NG2 mural cell loss, vascular pathology, and epileptiform electroencephalogram activity	[28]
7	TLE with or without HS, FCD	Humans	Pericyte-microglia assemblies with IBA1/HLA microglial cells outlining the capillary wall were observed in TLE-HS and FCD-IIb specimens Proinflammatory cytokines such as IL-1β cause morphological changes and IL-6 causes cell damage in human-derived pericytes	[29]
	NG2DsRed/C57BL6 (unilateral intra-hippocampal KA injections)	Mice	IL-1β elicited pericyte morphological changes and pericyte-microglia clustering in NG2DsRed hippocampal slices	
8	NG2DsRed/C57BL6 (unilateral intra-hippocampal KA injections)	Mice	Multicellular scarring occurs at the outer capillary wall in the hippocampus during seizure progression PDGFRβ stromal cells and collagens III and IV participate in the localized pericyte-glial scarring and capillary pathology in hippocampal subregions PDGFRβ is a proposed anti-inflammatory entry point for chronic disease stages in vivo	[30]
9	Transgenic mice (4-aminopyridine or low-Mg^2+^ conditions)	Mice	Pericytes regulate changes in vascular diameter in response to neuronal activity Recurrent seizures are associated with impaired neurovascular coupling and increased BBB permeability in capillaries Recurrent seizures lead to depolarization of pericytic mitochondria and subsequent vasoconstriction	[31]
10	Traumatic brain injury model (C57BL/6J mice with CCI and pilocarpine injections)	Mice	PDGFRβ levels were increased from 1 h to 4 days after CCI in the injured ipsilateral hippocampus prior to increased expression of markers of microglia and astrocytes; this supports the postulated role of pericytes as initiators of the CNS immune response Treatment with imatinib on postoperative days 0–4 reduced seizure susceptibility, demonstrating the usefulness of imatinib in vitro	[32]

CCI, controlled cortical impact; FCD, focal cortical dysplasia, HS, hippocampal sclerosis; IP, intraperitoneal; KA, kainic acid; PDGF-BB, platelet-derived growth factor-beta subunit; SE, status epilepticus; TBI, traumatic brain injury; TLE, temporal lobe epilepsy.

## Data Availability

The datasets generated and/or analyzed during the current study are available at the PubMed database repository (https://pubmed.ncbi.nlm.nih.gov/, accessed on 31 May 2021).

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
