# Peer review of "The Neuroinflammatory Role of Pericytes in Epilepsy"

_biomedicines, 2021, doi:10.3390/biomedicines9070759_

Round 1

Reviewer 1 Report

This review deals with the neuroinflammatory role of pericytes in epilepsy. The manuscript is well documented and cited essentially sufficient publications to point out the importance of the pericyte in epileptogenesis. Please consider the following minor points before final acceptance.

  1. Page 5, line 136, add space "tissue- and"
  2. Page 5, line 140, add MCP-1 to avoid a confusion as (MCP-1/CCL2)
  3. Page 6, line 187, prostaglandins (PGE2) should be prostaglandin E2 (PGE2) or prostaglandins (PGs)
  4. Page 7, line 208, add status epilepticus (SE) for the first appearance in the text.
  5. Page 7, line 214, status epilepticus should be SE
  6. Page 7, line 215, What's the DRE. Please spell out.
  7. Table 2, No.5, please use the abbreviation, TLE
  8. Table 2, No.6, please use the abbreviation, SE. "kainate" may be KA. And spell out EEG
  9. Table 2, No.9, "4-aminopyridine or", not "b4-aminopyridineor"
  10. Page 11, line 290, please spell out CA
  11. Page 12, line 322, please spell out OHC
  12. Page 12, line 323, It should be "by PDGF-BB (a PDGFRβ agonist) and imatinib (a PDGFβ inhibitor)
  13. Page 12, line 374, a slush seems better than a hyphen as IL-1/IL-1R1
  14. There are several publications relating to the topic. Please consider to add the following works.

Structural, Molecular, and Functional Alterations of the Blood-Brain Barrier during Epileptogenesis and Epilepsy: A Cause, Consequence, or Both?

Löscher W, Friedman A.

Int J Mol Sci. 2020 Jan 16;21(2):591. doi: 10.3390/ijms21020591.

High Mobility Group Box-1 and Blood-Brain Barrier Disruption.

Nishibori M, Wang D, Ousaka D, Wake H.

Cells. 2020 Dec 10;9(12):2650. doi: 10.3390/cells9122650.

Blood-Brain Barrier Dysfunction in Idiopathic Intracranial Hypertension.

Hasan-Olive MM, Hansson HA, Enger R, Nagelhus EA, Eide PK.

J Neuropathol Exp Neurol. 2019 Sep 1;78(9):808-818. doi: 10.1093/jnen/nlz063.

Blood-brain barrier leakage of blood proteins in idiopathic normal pressure hydrocephalus.

Eide PK, Hansson HA.

Brain Res. 2020 Jan 15;1727:146547. doi: 10.1016/j.brainres.2019.146547. Epub 2019 Nov 8.

Simvastatin, edaravone and dexamethasone protect against kainate-induced brain endothelial cell damage.

Barna L, Walter FR, Harazin A, Bocsik A, Kincses A, Tubak V, Jósvay K, Zvara Á, Campos-Bedolla P, Deli MA.

Fluids Barriers CNS. 2020 Feb 10;17(1):5. doi: 10.1186/s12987-019-0166-1.

Spatiotemporal dynamics of PDGFRbeta expression in pericytes and glial scar formation in penetrating brain injuries in adults.

Reeves C, Pradim-Jardim A, Sisodiya SM, Thom M, Liu JYW.

Neuropathol Appl Neurobiol. 2019 Oct;45(6):609-627. doi: 10.1111/nan.12539. Epub 2019 Apr 2.

Blood-brain barrier dysfunction as a potential therapeutic target for neurodegenerative disorders.

Uprety A, Kang Y, Kim SY.

Arch Pharm Res. 2021 May;44(5):487-498. doi: 10.1007/s12272-021-01332-8. Epub 2021 May 24.

Author Response

To the Reviewer 1

We appreciate the time and effort you have dedicated to providing this insightful feedback. We have revised our manuscript in accordance with your comments as much as possible. All suggested revisions and additional changes made to improve the language of the manuscript are indicated in underlined texts. We hope that, with these revisions, our manuscript will be suitable for publication in Biomedicines.

Responses to Reviewer 1’s comments

Comment 1

This review deals with the neuroinflammatory role of pericytes in epilepsy. The manuscript is well documented and cited essentially sufficient publications to point out the importance of the pericyte in epileptogenesis. Please consider the following minor points before final acceptance.

  1. Page 5, line 136, add space "tissue- and"
  2. Page 5, line 140, add MCP-1 to avoid a confusion as (MCP-1/CCL2)
  3. Page 6, line 187, prostaglandins (PGE2) should be prostaglandin E2 (PGE2) or prostaglandins (PGs)
  4. Page 7, line 208, add status epilepticus (SE) for the first appearance in the text.
  5. Page 7, line 214, status epilepticus should be SE
  6. Page 7, line 215, What's the DRE. Please spell out. drug-resistant epilepsy (DRE)
  7. Table 2, No.5, please use the abbreviation, TLE
  8. Table 2, No.6, please use the abbreviation, SE. "kainate" may be KA. And spell out EEG
  9. Table 2, No.9, "4-aminopyridine or", not "b4-aminopyridineor"
  10. Page 11, line 290, please spell out CA  cornu ammonis
  11. Page 12, line 322, please spell out OHC  organotypic hippocampal cultures
  12. Page 12, line 323, It should be "by PDGF-BB (a PDGFRβ agonist) and imatinib (a PDGFβ inhibitor)
  13. Page 12, line 374, a slush seems better than a hyphen as IL-1/IL-1R1
  14. There are several publications relating to the topic. Please consider to add the following works.

Response: Thank you for pointing this out. We have made the correction, according to your instruction.

Thank you for sharing your valuable paper with us. We have cited some of the papers under the below.

Structural, Molecular, and Functional Alterations of the Blood-Brain Barrier during Epileptogenesis and Epilepsy: A Cause, Consequence, or Both? 

Löscher W, Friedman A.

Int J Mol Sci. 2020 Jan 16;21(2):591. doi: 10.3390/ijms21020591.

High Mobility Group Box-1 and Blood-Brain Barrier Disruption.

Nishibori M, Wang D, Ousaka D, Wake H.

Cells. 2020 Dec 10;9(12):2650. doi: 10.3390/cells9122650.

Spatiotemporal dynamics of PDGFR beta expression in pericytes and glial scar formation in penetrating brain injuries in adults.

Reeves C, Pradim-Jardim A, Sisodiya SM, Thom M, Liu JYW.

Neuropathol Appl Neurobiol. 2019 Oct;45(6):609-627. doi: 10.1111/nan.12539. Epub 2019 Apr 2.

Blood-brain barrier dysfunction as a potential therapeutic target for neurodegenerative disorders.

Uprety A, Kang Y, Kim SY.

Arch Pharm Res. 2021 May;44(5):487-498. doi: 10.1007/s12272-021-01332-8. Epub 2021 May 24.

Reviewer 2 Report

In various portions of the review, the authors utilized "we" to promote/highlight specific work of the authors.  All such indications should be removed as they demonstrate bias and self advertisement. such concern is also raised on line 62 when suggesting highlights of novel directions of research, which leaves the reader under the impression the review was written with intent to self promote.

The review, on other criteria, is well written and appears to follow logical sequencing and train of thought.

This appears to be a more expanded topic reveiw following information previously submitted in the following item: Yamanaka, G.; Morichi, S.; Takamatsu, T.; Watanabe, Y.; Suzuki, S.; Ishida, Y.; Oana, S.; Yamazaki, T.; Takata, F.; Kawashima, H. Links between Immune Cells from the Periphery and the Brain in the Pathogenesis of Epilepsy: A Narrative Review. Int. J. Mol. Sci. 2021, 22, 4395.   doi: 10.1007/s00401-018-1893-0.  (properly cited)

as well as 

Acta Neuropathologica (2018) 136:507–523,  https://doi.org/10.1007/s00401-018-1893-0

(not cited but probably should be cited.)

It is a reasonable expansion of the work and I do not consider it to be inappropriate.  

Suggestion: In a new paragraph, the authors should include some previous reviews concerning pericytes relationship to neurological disorders and how this review differs and expands the knowledge base.

line 384  introduces a problem that should have further comment by the authors.  Concerning TGFb1, please complete the thought and concern suggested.    

I enjoyed reading this work and would be very happy to see it published when modified.

Author Response

To the Reviewer 2

We appreciate the time and effort you have dedicated to providing this insightful feedback. We have revised our manuscript in accordance with your comments as much as possible. However, we have limited time and may not have been able to make the corrections you mentioned. All suggested revisions, as well as additional sentences to improve the language of the manuscript, are indicated in underlined texts. We hope that, with these revisions, our manuscript will be suitable for publication in Biomedicines.

Responses to Reviewer 2’s comments

Comment 1.

In various portions of the review, the authors utilized "we" to promote/highlight specific work of the authors.  All such indications should be removed as they demonstrate bias and self advertisement. such concern is also raised on line 62 when suggesting highlights? of novel directions of research, which leaves the reader under the impression the review was written with intent to self promote. The review, on other criteria, is well written and appears to follow logical sequencing and train of thought.

Response: We deeply appreciate this suggestion and completely agree. Therefore, we have changed the sentence. We have revised the manuscript as much as possible according to your suggestion, but please let us know again if there is anything else you want us to do.

P.1 Line 39

we and others have demonstrated that brain pericytes respond to inflammatory signals, such as circulating cytokines

to

brain pericytes can respond to inflammatory signals, such as circulating cytokines

P.1 Line 41

We have recently demonstrated that pericytes may act as sensors for the inflammatory response in the CNS,

To

Recent studies have demonstrated that pericytes may act as sensors for the inflammatory response in the CNS,

P.5 Line 158

We have previously demonstrated that pericytes was more sensitive to pro-inflammatory cytokines compared to other cells in the NVU. [9, 11-13]

To

Pericytes have been shown to be more sensitive to pro-inflammatory cytokines compared to other cells in the NVU. [9, 11-13]

P.9 Line 234

Our analysis of pediatric patients with DRE also demonstrated a correlation between the number of seizures and intracellular IL-1β levels in monocytes

To

Current analysis of pediatric patients with DRE also demonstrated a correlation between the number of seizures and intracellular IL-1β levels in monocytes

P.12 Line 337

we have previously assessed how pericytes contribute to the pathogenesis of epilepsy using a controlled cortical impact (CCI) device,

to

The evaluation of the involvement of pericytes in the pathogenesis of epilepsy was performed using a controlled cortical impact (CCI) device. 

Comment 2.

This appears to be a more expanded topic reveiw following information previously submitted in the following item: Yamanaka, G.; Morichi, S.; Takamatsu, T.; Watanabe, Y.; Suzuki, S.; Ishida, Y.; Oana, S.; Yamazaki, T.; Takata, F.; Kawashima, H. Links between Immune Cells from the Periphery and the Brain in the Pathogenesis of Epilepsy: A Narrative Review. Int. J. Mol. Sci. 2021, 22, 4395.   doi: 10.1007/s00401-018-1893-0.  (properly cited)

as well as 

Acta Neuropathologica (2018) 136:507–523,  https://doi.org/10.1007/s00401-018-1893-0

Targeting pericytes for therapeutic approaches to neurological disorders

Jinping Cheng, Acta Neuropathologica volume 136, pages507–523 (2018) (not cited but probably should be cited.)

It is a reasonable expansion of the work and I do not consider it to be inappropriate.  

Suggestion: In a new paragraph, the authors should include some previous reviews concerning pericytes relationship to neurological disorders and how this review differs and expands the knowledge base.

Response: As pointed out, the present review is an update of the 2021 article (Int. J. Mol. Sci. 2021, 22, 4395.   doi: 10.1007/s00401-018-1893-0) and focuses on pericytes and the pathogenesis of epilepsy. In this review, we not only look at the link between the peripheral and central nervous system, but also at the relationship between epilepsy and pericytes from the perspective of neuroinflammation. Although there have been many reviews of pericytes, none have focused on epilepsy and pericytes, which is the reason for this review. We are very grateful for your valuable suggestions. We have inserted the following text, which you might like to see

P1. Line 58-64

The association between pericytes and epilepsy has attracted attention, while several recent studies have illustrated the contributions of pericytes to the pathogenesis of epilepsy. [2, 25-32] These studies suggested that pericytes might participate in the pathogenesis of epilepsy, consisting of neuroinflammation and BBB damage and the interaction between peripheral and central immunity. Thus, evidence on the relationship between pericytes and the pathogenesis of epilepsy is gradually accumulating. Therefore, this study aimed to investigate the pathogenesis of epilepsy and pericytes because none of the review articles focused on this, even though therapeutic targets for pericytes in neurological disorders were investigated [17, 33, 34].

Comment 3

line 384  introduces a problem that should have further comment by the authors.  Concerning TGFb1, please complete the thought and concern suggested.    

Response: We appreciate this suggestion. The following sentences have been added to the manuscript.

P13. Line 391-396

Since TGFβ1 suppresses pericyte phagocytosis and reduces the expression of central leukocyte trafficking chemokines and adhesion molecules while increasing the expression of pro-inflammatory cytokines and enzymes that promote BBB disruption, a paradoxical reaction has been reported. [27] The TGFβ1 response of pericytes may differ from the anti-inflammatory response of microglia[104-107]; therefore, further studies are required to obtain any effect on this non-uniform response.

Comment 4.

I enjoyed reading this work and would be very happy to see it published when modified.

Response: Thank you for your kind words. We have tried our best to meet your expectations. Please consider our revisions.

Reviewer 3 Report

In this manuscript, Gaku Yamanaka et al. have summarized the neuroinflammatory role of pericytes in epilepsy. The manuscript has discussed multiple aspects of pericytes involved in epilepsy with most updated evidence, which definitely will contribute to the research field of epilepsy. Overall, it is a well-written review article with higher quality and informatic figures. A minor suggestion for the manuscript is as following: the section 3 and 4 could be combined as one section “pericytes and neuroinflammation”; the section 5 and 6 should be after the section 7 because both the section 5 and 6 are providing evidence to indicate the pericyte participating in the epilepsy.

Author Response

To the Reviewer 3

We appreciate the time and effort you have dedicated to providing this insightful feedback. We have revised our manuscript in accordance with your comments. We hope that, with these revisions, our manuscript will be suitable for publication in Biomedicines.

Comment 1

In this manuscript, Gaku Yamanaka et al. have summarized the neuroinflammatory role of pericytes in epilepsy. The manuscript has discussed multiple aspects of pericytes involved in epilepsy with most updated evidence, which definitely will contribute to the research field of epilepsy. Overall, it is a well-written review article with higher quality and informatic figures. A minor suggestion for the manuscript is as following: the section 3 and 4 could be combined as one section “pericytes and neuroinflammation”; the section 5 and 6 should be after the section 7 because both the section 5 and 6 are providing evidence to indicate the pericyte participating in the epilepsy.

Response: Thank you for pointing this out. We have made the correction, according to your instruction.

Round 2

Reviewer 2 Report

Thank you for the changes.